# Changes in Outcomes of Macular Optical Coherence Tomography Angiography Following Surgery for Optic Disc Pit Maculopathy

**DOI:** 10.3390/diagnostics14090874

**Published:** 2024-04-23

**Authors:** Melih Akıdan, Muhammet Kazım Erol, Birumut Gedik, Mehmet Erkan Doğan, Ibrahim Başol, Elçin Süren

**Affiliations:** 1Department of Ophthalmology, Antalya Akseki State Hospital, 07630 Antalya, Turkey; 2Department of Ophthalmology, Antalya Training and Research Hospital, University of Health Sciences, 07100 Antalya, Turkey; muhammetkazimerol@gmail.com (M.K.E.); elcin_baskan@yahoo.com (E.S.); 3Department of Ophthalmology, Antalya Serik State Hospital, 07500 Antalya, Turkey; birumut.gedik@gmail.com; 4Department of Ophthalmology, Akdeniz University, 07059 Antalya, Turkey; m.erkandogan@gmail.com; 5Department of Ophthalmology, Antalya City Hospital, 07100 Antalya, Turkey; dr.basol@gmail.com

**Keywords:** choroidal blood flow, optical coherence tomography angiography, optic disc pit maculopathy, pars plana vitrectomy

## Abstract

Purpose: we aimed to report on the optical coherence tomography angiography (OCTA) outcomes of eight patients with optic disc pit maculopathy (ODP-M) who were treated with 23-gauge pars plana vitrectomy (PPV). Methods: We examined sixteen eyes of eight patients—eight eyes with ODP-M and eight healthy fellow eyes. Fundus color photography, fundus autofluorescence, fundus fluorescein angiography, optical coherence tomography (OCT), and OCTA were performed. The vascular density, choriocapillaris blood flow (CCBF), and foveal avascular zone (FAZ) were analyzed using OCTA. Moreover, the correlation between the best-corrected visual acuity (BCVA) and macular OCTA parameters was assessed. Results: Compared with the healthy fellow eyes, the eyes with ODP-M preoperatively were found to have decreased BCVA, superficial capillary plexus (SCP) vascular density (i.e., total, foveal, parafoveal, and perifoveal), deep capillary plexus (DCP) vascular density (i.e., total, parafoveal, and perifoveal), and CCBF but a significantly increased FAZ (*p* < 0.05). When the eyes with ODP-M were analyzed pre- and postoperatively at month 12 after surgery, the BCVA, SCP vascular density (i.e., perifoveal), and CCBF had significantly increased, and the FAZ had significantly decreased (*p* < 0.05). When the eyes with ODP-M were compared with the healthy fellow eyes postoperatively at month 12, the BCVA, SCP, and DCP vascular density parameters had increased, along with CCBF, and the FAZ had decreased in eyes with ODP-M, though not to the levels of the healthy fellow eyes (*p* < 0.05). Moreover, a positive correlation was found between the postoperative BCVA and SCP total vascular density (*p* < 0.05). Conclusion: The BCVA and macular OCTA parameters improved in eyes with ODP-M at month 12 following surgery. However, the BCVA and OCTA of the eyes operated on did not reach the levels of the healthy fellow eyes, possibly due to impaired choroidal blood flow (CBF) recovery and the presence of a larger FAZ. In summary, OCTA seems to be useful for assessing qualitative and quantitative perioperative microvascular changes.

## 1. Introduction

Optic disc pit (ODP), a rare congenital anomaly of the optic nerve head, is characterized by herniation of the dysplastic retinal tissue into an excavation that is rich in collagen and can extend into the subarachnoid space via a defect in the lamina cribrosa [1,2]. Although ODP is usually asymptomatic, 25% to 75% of patients develop serous retinal detachment and/or retinoschisis of the central macula in some stage in their lives, which results in so-called optic disc pit maculopathy (ODP-M) [2,3]. Cases of ODP-M may progress into lamellar or full-thickness macular hole (MH), cystic degeneration, and retina pigment epithelium (RPE) atrophy, which may lead to the irreversible deterioration in the best-corrected visual acuity (BCVA) [2,4].

Studies involving optical coherence tomography (OCT) have contributed considerably to determining the pathogenesis of ODP-M [5,6,7], detecting maculopathy-associated pathologies [8,9], surgical decision making, assessing responses to surgery, and identifying visual prognostic factors [4,10,11]. Despite these benefits, many controversies around the use of OCT remain.

Optical coherence tomography angiography (OCTA) is a functional extension of OCT that allows for the visualization and quantification of blood flow in the retinal and choroidal vasculature. OCTA generates depth-resolved images of the microvasculature by detecting motion contrast from flowing blood cells, without the need for dye injection. Although OCTA does provide three-dimensional volumetric data, it is more accurate to describe it as a non-invasive imaging technique that generates high-resolution depth-resolved images of the retinal and choroidal vasculature [12].

Through the use of OCTA, changes in the superficial capillary plexus (SCP) and deep capillary plexus (DCP), vascular density, and the foveal avascular zone (FAZ) can be assessed. OCTA has been demonstrated to be clinically useful in evaluating numerous retinal and choroidal conditions such as diabetic retinopathy, retinal vascular occlusions (RVOs), macular telangiectasia, choroidal neovascularisation (CNV), and inflammatory conditions [13].

There are only a few published OCTA studies on ODP-M. Apart from the study by Michalewska et al., most of the research involving OCTA in ODP-M has been limited to case reports primarily focusing on OCTA-based assessments of the optic disc. To the best of our knowledge, the study presented herein is the first to assess other macular OCTA parameters such as the vascular density, choriocapillaris blood flow (CCBF), and the FAZ. We believe that our findings can contribute to the literature on assessing the efficacy of surgical treatment, the postoperative follow-up of retinal vascularity, the clarification of visual outcomes, and the determination of the visual prognosis.

## 2. Materials and Methods

Approval for this study was obtained from the local ethics committee of the University of Health Sciences, Antalya Training and Research Hospital (approval date—number: 25 March 2024 and 070-2024). This study was conducted and performed in compliance with the ethical standards set out in the Declaration of Helsinki. This single-center retrospective study included patients who had undergone ODP-M surgery between 2017 and 2022. 

The inclusion criteria included patients who had undergone pars plana vitrectomy (PPV) due to a progressive increase in serous retinal detachment associated with ODP-M and worsening BCVA. The exclusion criteria included a refractive error >±6 spherical equivalent, an axial length <22 mm and >26 mm, and an intraocular pressure (IOP) >21 mmHg; an image signal strength index <60 to media opacity, motion, or other artifacts; pre-existing macular and optic disc pathologies, such as epiretinal membrane, macular hole, retinal vascular occlusion, diabetic retinopathy, glaucoma, optic neuropathy, or uveitis; prior ocular laser therapy, or vitreoretinal or other eye surgeries; and the presence of systemic disease and the use of medication.

All patients underwent a detailed ophthalmological examination, including measurements of the best-corrected visual acuity (BCVA) using a Snellen chart, a biomicroscopic examination of the anterior segment, intraocular pressure measurement with Goldmann applanation tonometry, dilated fundus examination, OCTA imaging (RTVue XR100-2, Optovue, Fremont, CA, USA), and OCT (RTvue100, Optovue, Fremont, CA, USA). Each patient with ODP-M underwent fundus color photography, fundus autofluorescence, and fundus fluorescein angiography (Visucam NM/FA, Carl Zeiss, Germany) to confirm the diagnosis.

All measurements were recorded before and 1 week, 3 months, 6 months, and 12 months after 23-gauge PPV surgery.

### 2.1. OCTA Measurements

Amplitude-decorrelation angiographic images were obtained using an OCTA device. For the imaging of all patients, a commercial OCTA device (RTVue XR100-2, Optovue, Fremont, CA, USA) with a scan rate of 70,000 A-scans/s, a scan beam wavelength of 840 ± 10 nm, and a bandwidth of 45 nm was used. With this device, the volumetric scanning of 304 × 304 A-scans can be performed at a rate of 70,000 A-scans/s in approximately 3.0 s. Afterward, 6 × 6 mm OCT angiogram software (Version Phase 7) was used to evaluate the vascular structures. 

The software (Version Phase 7) automatically segmented full-thickness retinal scans into the SCP and DCP, outer retina, and choriocapillaris. It also automatically calculated the vascular density in the SCP and DCP vascular zone, which in turn automatically revealed the FAZ area. The starting line of the SCP was set as the area 3 μm beneath the internal limiting membrane (ILM), where the thickness was sufficient to contain the ganglion cell layer and the capillaries surrounding the FAZ in the central macular region. Imaging of the DCP began 15 μm beneath the outer border of the inner plexiform layer and ended 70 μm beneath the inner plexiform layer. 

Foveal vascular density was determined as the percentage of vessel density in a 1 mm circle centered on the fovea, total vascular density was determined as the percentage of vessel density in a 6 mm circle centered on the fovea, parafoveal vascular density was determined as the percentage of vessel density in a ring-shaped area between 1 mm and 3 mm, and perifoveal vascular density was determined as the percentage of vessel density in a ring-shaped area between 3 mm and 6 mm. These density measurements were obtained cross-sectionally using the automatic mode of the device. The ratio of the vascular image in these areas to the whole area provided the density value as a percentage.

The FAZ, defined as the avascular area in the center of the fovea, was measured. It was analyzed as a single layer and not separated into the SCP and DCP. The same software was used to calculate the blood flow in a central circular zone of 3.144 mm^2^ in the outer retina and choriocapillaris segments. 

The presence of preoperative significant serous retinal detachment might limit the reliability of OCTA measurements; to combat this issue, a single trained reader (M.A.) segmented the OCTA sections manually and compared them with the automatic measurements. No patients were found to have major segmentation errors. Moreover, the result of the measurements with an image signal strength index greater than 60 was included in this study to eliminate the likelihood of inaccurate measurements due to artefacts.

Although 6 × 6 mm OCTA disc imaging was performed on all patients, the results could not be included in the final evaluation, because the device did not provide numerical data.

### 2.2. Surgery

All surgical procedures were performed using a Leica F19 surgical microscope and Constellation Vitrectomy System (Version 5.01.08, Alcon, Fort Worth, TX, USA). Retrobulbar anesthesia was administered by an experienced surgeon (M.K.E.) during all surgeries. In all eyes, 23-gauge PPV was performed with the triamcinolone-assisted removal of the posterior hyaloid interface. The ILM was stained with a 0.05% solution of brilliant blue (Ocublu, Miray Medical, Bursa/Turkey) dye for approximately 30 s. The ILM was subsequently grasped at a distance of 2 disc diameters from the fovea and peeled around the fovea in a circular manner. The center of the fovea was not subjected to ILM peeling, which involves fovea-sparing ILM peeling with the use of an inverted flap technique. Next, the peeled ILM was inverted and flattened on the optic disc such that it covered the pit. Following fovea-sparing ILM peeling with the use of an inverted flap technique, fluid–air exchange and endodrainage were performed through the pit using a silicone-tipped back flush needle. One or two rounds of barrage endolaser treatment were performed. Finally, 14% C3F8 was injected into the vitreous to stabilize the ILM flap over the pit area. During the postoperative period, patients were instructed to maintain a face-down position for 3 days. Antibiotic, steroidal, and non-steroidal anti-inflammatory eye drops were prescribed for 4 weeks for all patients.

A full-thickness MH was detected in two patients at postoperative month 3. One patient was reoperated on due to decreased vision that was considered to have not recovered spontaneously (i.e., MH size > 400 μm). In this patient’s case, wider ILM peeling that also included the fovea was performed using an inverted ILM flap technique, and the patient’s MH closed during follow-up (Figure 1 and Figure 2). None of the patients underwent cataract surgery simultaneously with or following PPV within the 1-year follow-up period. 

### 2.3. Statistical Analysis

The Statistical Package for Social Sciences for Windows version 23.0 (SPSS Inc., Chicago, IL, USA) was used for all statistical analyses. Continuous variables were recorded as the mean (M) ± standard deviation (SD), while categorical data were recorded as frequencies. Continuous variables were compared between the groups (i.e., eyes with ODP-M vs. healthy fellow eyes) by applying the independent samples *t*-test and the Mann–Whitney *U* test. Spearman’s correlation coefficient was also used to analyze the correlation between the continuous variables. A *p*-value of <0.05 was considered to indicate statistical significance.

## 3. Results

In this study, we examined sixteen eyes (i.e., eight eyes with ODP-M and eight healthy fellow eyes) of eight patients—five males and three females. The mean age of the patients was 37.1 years (range: 18–50 years). Of the eight eyes that were operated on, five were right eyes and three were left eyes. 

The preoperative OCT findings revealed that four patients had intraretinal fluid (IRF), subretinal fluid (SRF), and an outer lamellar hole (OLH) concomitantly; concurrently, three patients had IRF and SRF, and one patient had only IRF (Table 1). 

The BCVA and OCTA parameters of the eyes with ODP-M and healthy fellow eyes were compared preoperatively (Table 2). BCVA was significantly lower in the eyes with ODP-M (0.06 ± 0.02, *p* = 0.003). SCP total (43.95 ± 3.55, *p* = 0.004), foveal (23.28 ± 1.80, *p* = 0.004), parafoveal (46.46 ± 4.01, *p* = 0.01), and perifoveal (47.16 ± 0.93, *p* = 0.004) vascular densities were significantly lower in eyes with ODP-M. DCP total (46.98 ± 4.25, *p* = 0.006), parafoveal (50.96 ± 2.30, *p* = 0.004), and perifoveal (47.53 ± 6.06, *p* = 0.02) vascular densities were significantly lower in the eyes with ODP-M. The FAZ was significantly larger in the eyes with ODP-M (0.32 ± 0.04, *p* = 0.004). CCBF, however, was significantly reduced in the eyes with ODP-M (1.78 ± 0.31, *p* = 0.004).

A preoperative and postoperative (i.e., at month 12) comparison of the eyes with ODP-M was performed (Table 3). Postoperative BCVA (0.50 ± 0.17, *p* = 0.004) was found to have increased significantly, along with SCP perifoveal (49.33 ± 1.11, *p* = 0.01) vascular density and CCBF (2.09 ± 0.15, *p* = 0.02). However, the FAZ (0.26 ± 0.03, *p* = 0.02) was found to have significantly decreased. 

The eyes with ODP-M were compared with the healthy fellow eyes at month 12 following surgery (Table 4). BCVA was significantly higher (0.92 ± 0.27, *p* = 0.03) in the healthy fellow eyes. They also had significantly higher SCP total (52.15 ± 1.86, *p* = 0.01), foveal (28.60 ± 1.19, *p* = 0.004), and perifoveal (52.90 ± 2.04, *p* = 0.01) vascular densities. DCP total (56.05 ± 2.31, *p* = 0.01) and parafoveal (58 ± 2.21, *p* = 0.004) vascular densities were also significantly higher. Although CCBF was significantly higher as well (2.39 ± 0.30, *p* = 0.02), this was also the case for the FAZ (0.26 ± 0.03, *p* = 0.004) in this patient group.

The correlation of the BCVA and OCTA parameters of the eyes with ODP-M was analyzed postoperatively (Table 5). Only SCP total vascular density and BCVA were found to have a significantly positive correlation (*p* = 0.04).

## 4. Discussion

In this study, we evaluated perioperative changes in BCVA and macular OCTA parameters in eyes with ODP-M. At the postoperative 12-month follow-up, the eyes with ODP-M showed improvement in the vascular density parameters, as well as increased CCBF and decreased FAZ. Both preoperative and postoperative 12-month comparisons between the eyes with ODP-M and the healthy fellow eyes demonstrated that the BCVA and OCTA values of the eyes with ODP-M had not reached the values of their healthy fellow eyes. Moreover, a positive correlation was found between postoperative BCVA and SCP total vascular density. 

The literature contains very few studies involving the use of OCTA for ODP-M. In one study, Michalewska et al. assigned a group of 15 patients to Group 1 (i.e., SRF, with or without IRF) and Group 2 (i.e., IRF, with or without OLH) and compared only the FAZ of all OCTA parameters with the perioperative values [14]. They found that the FAZ increased after surgery in Group 2; however, no significant differences were found in the final BCVA between the groups. They concluded that the increased FAZ may have been related to prolonged postoperative anatomic recovery or altered retinal geometry in the FAZ. In our study, because there was an insufficient number of patients, we could not divide them into similar groups according to the preoperative OCT findings and therefore statistically compare the perioperative BCVA and OCTA parameters between the groups.

The association of the FAZ and vascular density with BCVA has been demonstrated a number of times through the use of OCTA. In particular, the association of vascular density and the FAZ with the severity of predictive diabetic retinopathy and BCVA in individuals with diabetic retinopathy has been analyzed in numerous studies [15,16]. Similar reports on other retinal vascular diseases, including ocular ischemic syndrome and RVO, have also been published [17,18]. Bonfiglio et al. treated rhegmatogenous retinal detachment (RRD) with PPV and reported that BCVA was associated with the FAZ and vascular density [19]. In a similar manner, in their study on RRD, Ng et al. found that a smaller FAZ in the DCP was associated with better visual acuity [20]. Woo et al., in contrast, identified a significantly negative correlation between postoperative BCVA and the superficial and deep FAZ in patients with the macula-off RRD group [21]. In another study, Wilczynski et al. performed surgery for full-thickness MH and found that the preoperative FAZ was correlated with postoperative BCVA, while Tsuboi et al. proposed that postoperative changes in the FAZ were correlated with visual improvement [22,23]. In a study involving surgery for MH patients, Chen et al. concluded that a greater reduction in the FAZ was related to better recovery of the central retinal tissue and a greater improvement in visual function [24].

Choroidal circulation plays an important role in nourishing the outer retina, particularly the photoreceptor layers, and several studies have demonstrated CBF’s central role in visual function [25]. In a study involving patients with RVO, Arıbas et al. found that reduced CBF played a role in the impairment of the outer retina (i.e., the ellipsoid, external limiting membrane, and retina pigment epithelium) and contributed to poor vision [26]. In contrast, Abbouda et al., finding a negative correlation between CBF and the FAZ, argued that reduced CBF might play a role in deteriorated visual function due to the enlargement of the FAZ [27]. In another study involving patients who underwent surgery for RRD, Çetinkaya Yaprak et al. found a larger FAZ and decreased CBF compared with healthy fellow eyes, in addition to a negative correlation between CBF and postoperative BCVA due to possible ischemia [28]. Wang et al., who performed surgery on patients with RRD, additionally demonstrated that CBF is positively correlated with BCVA at the 12-month follow-up [29]. They suggested that a follow-up on OCTA and CBF could help in explaining the differences in the patients’ postoperative visual outcomes. 

Even if the macula achieves complete anatomical flattening after surgery, it does not always regain full function. Among other possible reasons for this, blood perfusion restoration may take a much longer time [30]. At this point, the recovery of the impaired CBF and improvement in the outer retinal morphology, especially the ellipsoid zone structure, are considered to be the chief factors involved in the recovery time and BCVA [30]. Although recovery times vary, Sato et al. found, using a retinal flowmeter, that ocular microcirculation normalized at month 6 in patients operated on for RRD [31]. In a study including patients who underwent surgery for MHs, Gedik et al. showed, using OCTA, that CBF reached levels comparable to healthy fellow eyes at month 6 after surgery [32]. 

In our study, although we did not observe any correlation between BCVA and the FAZ or CBF and the FAZ, we did find a significantly positive correlation between BCVA and total SCP vascular density. At the 12-month follow-up, anatomical closure was achieved after surgery on the eyes with ODP-M, while an improvement in BCVA, increase in vascular density and CBF, and significant decrease in FAZ were observed after surgery. However, neither BCVA nor macular OCTA parameters of the eyes with ODP-M reached the values observed for the healthy fellow eyes. Such outcomes suggest that blood perfusion restoration and recovery time might take much longer in the eyes with ODP-M than the observations made in our study. In our study, full-thickness MH was detected in two patients at postoperative month 3. One patient was reoperated on due to poor vision, with the consideration that it would not recover spontaneously and that its closure had been achieved. In another patient, spontaneous closure was observed during the follow-up period. Remarkably, the improved BCVA and OCTA parameters in those two patients were at a lower level than those in the other patients at month 12. This difference, however, was not statistically significant. Prolonged reattachment time after surgery, delayed anatomical recovery, more impaired CBF, and a larger FAZ in these patients compared with other patients may explain these findings. We therefore caution that blood perfusion restoration may take far longer in these patients than in others. 

Most OCTA studies on ODP-M are case reports, except for the study by Michalewska et al., and include OCTA disc assessments. In a study conducted by Roizenblatt et al. on a 19-year-old patient with ODP-M, preoperative OCTA revealed that artefactual subfoveal CNV and postoperative OCTA were normal [33]. In the patients in our study, we did not observe an increased signal in either the en face image or the outer retina or choriocapillaris segment that would suggest perioperative CNV. Lutsenko et al. made the decision to perform surgery on a 27-year-old patient after OCTA findings revealed vascular density loss and visual deterioration. Following surgery, they observed, using OCTA, the increased vascular density of the SCP in the foveal and parafoveal regions and, in turn, suggested that OCTA is important in diagnosis, monitoring the course of the disease, and assessing the efficacy of surgical treatment [34]. Arturo et al. performed surgery on a 12-year-old patient, after which the OCTA findings revealed a fine texture of the choroid capillary layer and star-shaped folds corresponding to both the outer retina and the choroid capillary layer. The OCTA findings revealed that macular perfusion had normalized, which correlated with the full recovery of visual acuity [35]. In a 52-year-old patient with bilateral ODP without maculopathy, Chebil et al. used OCTA to reveal decreased vascular density with no flow within the defect [36]. However, OCTA parameters for the macula and disc were not measured. Similarly, Bach et al. described a 9-year-old asymptomatic patient with a radial spoke of capillary dropout emanating from the optic disc pit area using OCTA [37]. In contrast, Venkatesh et al. observed the absence of radial peripapillary capillaries in the affected area using OCTA in two asymptomatic patients with ODP and without maculopathy. They suggested that the subsequent loss of radial peripapillary capillaries could be due to the initial microstructural and microvascular changes prior to maculopathy and argued that using OCTA and central visual field defect detection could be a precursor of maculopathy [38]. In another study, Jiang et al. assessed the three layers of the optic nerve of eight eyes (i.e., four with ODP and four normal contralateral eyes) with OCTA disc assessments [39]. Although they detected a loss of vascular density and reduced vision in some areas of the eyes with ODP, they did not assess the macular parameters with OCTA. In contrast, we performed OCTA disc imaging for all patients; however, the images could not be included in our final evaluation, because the device did not provide numerical data.

Although several forms of treatment for ODP-M have been proposed, there is no consensus on the primary treatment for the persistence of macular detachment after the initial intervention. Reviews of various surgical combinations seem to favor the maximal approach because it combines vitrectomy, laser, and gas tamponade for optimal surgical outcomes [6,40]. There is also no overwhelming consensus as to whether combining PPV with ILM peeling can increase surgical success [41]. Although combining PPV with an inverted ILM flap and free ILM flap transplantation, autologous scleral flap, or platelets and using Tisseel fibrin sealant techniques have been associated with positive outcomes, Wagner et al. demonstrated that ILM peeling did not significantly improve vision [6,42,43]. The multi-center European VitreoRetinal Society (EVRS) Optic Disc Study has provided up-to-date information about the surgery and highlighted that ILM peeling is in fact unnecessary; on the contrary, it was found to increase the risk of developing MHs and was not associated with better vision even though anatomical closure was achieved in some cases of retreatment [44]. MHs are believed to develop during ILM peeling due to factors such as tangential and anteroposterior traction, the posterior hyaloid’s strong adherence to the thinned-out retina, and iatrogenic trauma [44,45]. In a case presentation, Roy et al. used a fovea-sparing ILM flap technique instead of conventional ILM peeling because the retina of the patient under examination was extremely thin and there was a risk of them developing postoperative MHs [46]. In turn, they demonstrated that the technique was beneficial for such patients. Similar to Roy et al., [46] we used a fovea-sparing ILM flap technique in our study and decided on retreatment surgery for an MH in one patient only, as we suspected that it might not close spontaneously. We believe that strong tractional forces could not be prevented and led to the development of the MH even though the foveal ILM was spared in this particular patient. 

Our study has some important limitations. One is the low number of patients included in this study, which we believe can be rectified by future studies involving more participants. Another limitation of our study is the likelihood that our results may be coincidental, since OCTA had several variables. OCTA segmentation errors and possible artefacts may affect the results of studies involving its use. We manually segmented the OCTA sections and compared them with the automatic measurements. We believe that we eliminated the likelihood of inaccurate measurements by ensuring that the image quality index was high. The third limitation of our study is that there are very few published studies reporting macular and disc OCTA results for eyes with ODP-M, meaning that we could not compare our results with past findings. 

## 5. Conclusions

In our study, changes in BCVA and OCTA parameters such as vascular density, CCBF, and the FAZ in eyes with ODP-M were compared perioperatively both with each other and with healthy fellow eyes. Although statistically significant improvements were observed after surgery in the eyes with ODP-M, they had not improved to the level of the healthy fellow eyes. We argue that this difference was caused by changes, especially in CBF and the FAZ, that occurred during blood restoration after successful surgery. We believe that OCTA can substantially contribute to knowledge about ODP-M in many ways, especially by aiding in the evaluation of qualitative and quantitative microvascular changes. In light of this, our findings stand to support future studies on ODP-M.

## Figures and Tables

**Figure 1 diagnostics-14-00874-f001:**
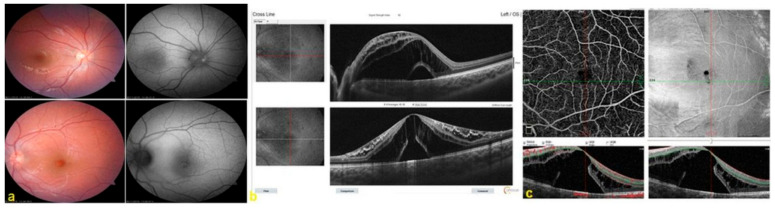
Preoperative images of the right and left eye of an 18-year-old female patient (Patient 1). (**a**) Fundus and fundus autofluorescence (FAF) images. (**b**) Optical coherence tomography (OCT) images of the left eye’s intraretinal (IRF) and subretinal fluid (SRF) with an outer lamellar hole (OLH). (**c**) Optical coherence tomography angiography (OCTA) superficial capillary plexus (SCP) cross-section (B-scan) and en face images of the left eye.

**Figure 2 diagnostics-14-00874-f002:**
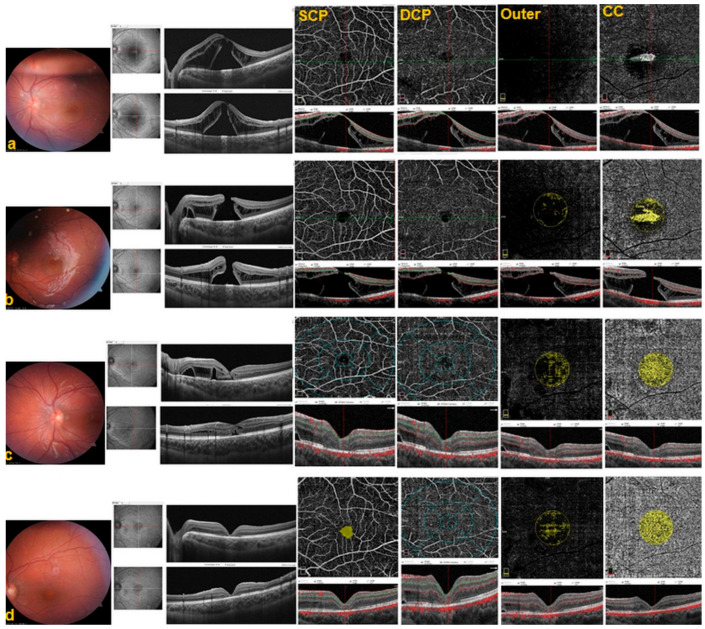
Postoperative images of the left eye of an 18-year-old female patient (Patient 1). (**a**) Image of the fundus 1 week after surgery; OCT images of persistent IRF and SRF with the OLH. OCTA cross-sectional (B-scan) images of the SCP, DCP, outer retina, and CC. (**b**) Image of the fundus at month 3 post surgery (before the second surgery); OCT images of the full-thickness MH. OCTA cross-section (B-scan) images of the SCP, DCP, outer retina, and CC. OCTA blood flow areas (yellow) of the outer retina and CC. (**c**) Image of the fundus at month 6 post surgery; OCT images of the resolved SRF, the closed MH, and cystic changes. OCTA vascular density measurement rings of the SCP and DCP. OCTA blood flow areas (yellow) of the outer retina and CC. (**d**) Image of the fundus at month 12 post surgery; OCT images of decreased cystic changes and continued IS/OS band restoration. OCTA FAZ area (yellow) of the SCP. OCTA blood flow areas (yellow) of the outer retina and CC.

**Table 1 diagnostics-14-00874-t001:** Demographic characteristics, BCVA, and preoperative OCT findings in eyes with ODP-M.

Patient	Age	Sex	Eye	BCVA-Preop	Preop-OCT	BCVA-Postop	Retreatment
1	18	F	L	0.04	IRF, SRF, OLH	0.32	+(MH)
2	23	F	L	0.1	IRF	0.4	-
3	38	M	L	0.05	IRF, SRF, OLH	0.5	-
4	50	M	L	0.04	IRF, SRF	0.63	−(MH)
5	42	F	L	0.1	IRF, SRF	0.63	-
6	45	M	L	0.05	IRF, SRF, OLH	0.5	-
7	48	M	R	0.1	IRF, SRF	0.63	-
8	33	M	L	0.032	IRF, SRF, OLH	0.4	-

F: female; M: male; L: left; R: right; BCVA: best-corrected visual acuity; OCT: optical coherence tomography; IRF: intraretinal fluid; SRF: subretinal fluid; OLH: outer lamellar hole; MH: macular hole.

**Table 2 diagnostics-14-00874-t002:** Preoperative comparison of the OCTA values of eyes with ODP-M and healthy fellow (control) eyes.

Variables	Group	Mean	S.D.	Min.	Max.	*p*
**BCVA**	ODP-M	0.06	0.02	0.032	0.10	**0.003 ***
	Control	0.92	0.27	0.8	1	
**SCP total vascular density (%)**	ODP-M	43.95	3.55	38.3	48.3	**0.004 ***
	Control	52.15	1.86	49	54.3	
**SCP foveal vascular density (%)**	ODP-M	23.28	1.80	19.9	24.9	**0.004 ***
	Control	28.60	1.19	26.6	30.1	
**SCP parafoveal vascular density (%)**	ODP-M	46.46	4.01	40.1	50.3	**0.01 ***
	Control	53.51	2.57	49.2	56.8	
**SCP perifoveal vascular density (%)**	ODP-M	47.16	0.93	45.9	48.3	**0.004 ***
	Control	52.90	2.04	49.7	55.6	
**DCP total vascular density (%)**	ODP-M	46.98	4.25	42.2	52.6	**0.006 ***
	Control	56.05	2.31	52	58.5	
**DCP foveal vascular density (%)**	ODP-M	33.16	5.42	27.2	40.7	0.055
	Control	41.65	6.80	32.8	50.3	
**DCP parafoveal vascular density (%)**	ODP-M	50.96	2.30	47.2	53.3	**0.004 ***
	Control	58	2.21	55.5	61.4	
**DCP perifoveal vascular density (%)**	ODP-M	47.53	6.06	40.8	55.2	**0.02 ***
	Control	56.60	3.24	51.3	59.6	
**Foveal avascular zone (mm^2^)**	ODP-M	0.32	0.04	0.26	0.38	**0.004 ***
	Control	0.19	0.01	0.16	0.21	
**Choriocapillaris blood flow (mm^2^)**	ODP-M	1.78	0.31	1.18	2.08	**0.004 ***
	Control	2.39	0.30	2.17	2.98	

BCVA: best-corrected visual acuity; S.D.: standard deviation; Min.: minimum; Max.: maximum; SCP: superficial capillary plexus; DCP: deep capillary plexus. * Mann–Whitney *U* test, significant at *p* < 0.05.

**Table 3 diagnostics-14-00874-t003:** Preoperative and month 12 postoperative comparison of the OCTA values of eyes with ODP-M.

Variables	Group	Mean	S.D.	Min.	Max.	*p*
**BCVA**	Preoperative	0.06	0.02	0.032	0.10	**0.004 ***
	Postoperative	0.50	0.17	0.32	0.63	
**SCP total vascular density (%)**	Preoperative	43.95	3.55	38.3	48.3	0.10
	Postoperative	47.15	2.92	42.5	50	
**SCP foveal vascular density (%)**	Preoperative	23.28	1.80	19.9	24.9	0.055
	Postoperative	25.05	1.58	22.4	26.8	
**SCP parafoveal vascular density (%)**	Preoperative	46.46	4.01	40.1	50.3	0.26
	Postoperative	49.06	4.52	42.2	54.9	
**SCP perifoveal vascular density (%)**	Preoperative	47.16	0.93	45.9	48.3	**0.01 ***
	Postoperative	49.33	1.11	47.5	50.6	
**DCP total vascular density (%)**	Preoperative	46.98	4.25	42.2	52.6	0.15
	Postoperative	50.71	3.68	44.4	54.4	
**DCP foveal vascular density (%)**	Preoperative	33.16	5.42	27.2	40.7	0.26
	Postoperative	37.48	7.08	30.1	47	
**DCP parafoveal vascular density (%)**	Preoperative	50.96	2.30	47.2	53.3	0.07
	Postoperative	53.53	1.82	51.2	56	
**DCP perifoveal vascular density (%)**	Preoperative	47.53	6.06	40.8	55.2	0.20
	Postoperative	52.20	3.99	45	55.8	
**Foveal avascular zone (mm^2^)**	Preoperative	0.32	0.04	0.26	0.38	**0.02 ***
	Postoperative	0.26	0.03	0.21	0.30	
**Choriocapillaris blood flow (mm^2^)**	Preoperative	1.78	0.31	1.18	2.08	**0.02 ***
	Postoperative	2.09	0.15	1.81	2.28	

BCVA: best-corrected visual acuity; S.D.: standard deviation; Min.: minimum; Max.: maximum; SCP: superficial capillary plexus; DCP: deep capillary plexus. * Mann–Whitney U test, significant at *p* < 0.05.

**Table 4 diagnostics-14-00874-t004:** Comparison of month 12 postoperative OCTA values of eyes with ODP-M and healthy fellow eyes.

Variables	Group	Mean	S.D.	Min.	Max.	*p*
**BCVA**	Control	0.92	0.27	0.8	1	**0.03 ***
	Postoperative	0.50	0.17	0.32	0.63	
**SCP total vascular density (%)**	Control	52.15	1.86	49	54.3	**0.01 ***
	Postoperative	47.15	2.92	42.5	50	
**SCP foveal vascular density (%)**	Control	28.60	1.19	26.6	30.1	**0.004 ***
	Postoperative	25.05	1.58	22.4	26.8	
**SCP parafoveal vascular density (%)**	Control	53.51	2.57	49.2	56.8	0.07
	Postoperative	49.06	4.52	42.2	54.9	
**SCP perifoveal vascular density (%)**	Control	52.90	2.04	49.7	55.6	**0.01 ***
	Postoperative	49.33	1.11	47.5	50.6	
**DCP total vascular density (%)**	Control	56.05	2.31	52	58.5	**0.01 ***
	Postoperative	50.71	3.68	44.4	54.4	
**DCP foveal vascular density (%)**	Control	41.65	6.80	32.8	50.3	0.20
	Postoperative	37.48	7.08	30.1	47	
**DCP parafoveal vascular density (%)**	Control	58	2.21	55.5	61.4	**0.004 ***
	Postoperative	53.53	1.82	51.2	56	
**DCP perifoveal vascular density (%)**	Control	56.60	3.24	51.3	59.6	0.055
	Postoperative	52.20	3.99	45	55.8	
**Foveal avascular zone (mm^2^)**	Control	0.19	0.01	0.16	0.21	**0.004 ***
	Postoperative	0.26	0.03	0.21	0.30	
**Choriocapillaris blood flow (mm^2^)**	Control	2.39	0.30	2.17	2.98	**0.02 ***
	Postoperative	2.09	0.15	1.81	2.28	

BCVA: best-corrected visual acuity; S.D.: standard deviation; Min.: minimum; Max.: maximum; SCP: superficial capillary plexus; DCP: deep capillary plexus. * Mann–Whitney U test, significant at *p* < 0.05.

**Table 5 diagnostics-14-00874-t005:** Correlation of the postoperative BCVA of the eyes with ODP-M and postoperative OCTA values.

Variables	r	*p*
**SCP total vascular density (%)**	0.820	**0.04 ***
**SCP foveal vascular density (%)**	0.273	0.60
**SCP parafoveal vascular density (%)**	0.213	0.68
**SCP perifoveal vascular density (%)**	0.213	0.68
**DCP total vascular density (%)**	0.091	0.86
**DCP foveal vascular density (%)**	0.395	0.43
**DCP parafoveal vascular density (%)**	0.213	0.68
**DCP perifoveal vascular density (%)**	0.030	0.95
**Foveal avascular zone (mm^2^)**	0.395	0.43
**Choriocapillaris blood flow (mm^2^)**	0.091	0.86

SCP: superficial capillary plexus; DCP: deep capillary plexus. * Spearman’s rank correlation, significant at *p* < 0.05.

## Data Availability

Data are available on request from the authors.

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
