# Peer review of "Changes in Outcomes of Macular Optical Coherence Tomography Angiography Following Surgery for Optic Disc Pit Maculopathy"

_diagnostics, 2024, doi:10.3390/diagnostics14090874_

Round 1

Reviewer 1 Report

Comments and Suggestions for Authors

Specific comments:

Abstract: There are inconsistency in punctuation, especially after colons ( : ) some places there is space provided while in others there is no space.

L18: Here the word subjected is not appropriate, either change just the term or the whole sentence.

L26, 29, 33: Month 12, write M in lower case.

L25-28: How did you know the changes (BCVA, SCP vascular density 26 (i.e. perifoveal) and CCBF had significantly increased and the FAZ had significantly decreased 27 (p<0.05)? Was the 12 month findings compared to preoperative data or some other data?

L36: The sentence seems incomplete – your intention might be to write “to be” instead of only “to”.

Overall, the abstract writing needs improvement in language, presentation, and punctuation.

Keywords:

Unfortunate that I see signs of carelessness throughout the manuscript. Even in keywords, punctuation, and words like planavitrectomy is written as a single word, which should be in two words.  

Secondly, it looks good if the keywords are arranged in an alphabetical order.

Introduction:

L44: Any reason for writing %25 to %75? Ideally, it is written as 25%.

L48: Correct grammar.

L53: Remove this redundant sentence – “Studies conducted with OCT thus shed light on ODP-M day by day.”

L58-60: Correct this sentence.

L64-72: This paragraph needs complete re-writing. Secondly, it is not clear why you have mentioned a study specifically out of many such studies or case reports. It looks odd.

Materials and Methods

L74-76: Re-write this sentence, possibly break it into 2 sentences.

L74: What were the inclusion criteria, and also the exclusion criteria?

L79-81: The follow up periods writing may be mentioned after the eye examination and imaging.

Please write the Methods section in sequence – ethical points, recruitment with inclusion and exclusion criteria, ophthalmic examination, imaging, and follow up.

L97: How accurate is this automatic segmentation?

Q: You have not mentioned which FAZ you have considered. SCP, MCP or DCP. Literature says the FAZ with MCP is the most consistent one.

Q: How did you differentiate CCBF and CBF?

L102, 116: You have defined ILM in these lines. Remove the later definition keeping only the abbreviation.

Q: What was the deciding factor for the size of ILM peeling?

Q: Can you defend yourself if your analysis of 16 eyes holds statistical significance for your study to be worth clinical application? Please include this paragraph in the manuscript.

Results

L155-156: “The 16 eyes (i.e. eight eyes with ODP-M and eight healthy fellow eyes) of eight patients—five male and three female— were included in the study.” You have used this exact sentence three times including in the abstract. You should remove it from the Methods section if you wish to write it in the Results.

Discussion:

Are you using CCBF and CBF interchangeably, for example in L258?

L274: This sentence is not complete.

L274-290: Many points are just repeated frequently in the manuscript. Correct this, and do not repeat the points.

L332-333: What are you trying to say here?

Over the manuscript, you have mention “In a study” 5 times.

Final suggestions:

1.      Improve language, grammar and presentation. Punctuations and spacing mistakes are seen throughout the manuscript. These are signs of carelessness. Professional language editing service may help.

2.      Avoid repeating points.

Comments on the Quality of English Language

Extensive editing required.

Reviewer 2 Report

Comments and Suggestions for Authors

This study provides valuable preliminary insights into the utility of OCTA for quantitative evaluation of macular vascular changes in eyes with ODP-M undergoing surgical treatment with pars plana vitrectomy. I have a few comments to make:

·      Segmentation errors: In eyes with ODP-M, the presence of intraretinal and subretinal fluid, retinoschisis, and alterations in retinal layering could potentially confound accurate segmentation of the superficial and deep capillary plexuses. This may lead to incorrect calculation of vascular density and other OCTA parameters. Did the authors observe any segmentation errors in the ODP-M eyes? If so, was manual correction of the segmentation performed? The authors should elaborate more on this, and if significant segmentation artifacts were present, the y should acknowledge this as a major limitation of their vascular density calculations.

·      In the methods section, please specify how did you define foveal, parafoveal, and perifoveal regions.

·      On page 2, lines 53-54, you mention the following: ‘Studies conducted with OCT thus shed light on ODP-M day by day. Even so, many controversies surrounding OCT remain.’ What do you mean by controversies here?

·      One page 2, line 55, you mention that OCTA is a three-dimensional OCT. OCTA is not a three-dimensional OCT; rather, it is a functional extension of OCT that allows for the visualization and quantification of blood flow in the retinal and choroidal vasculature. OCTA generates depth-resolved images of the microvasculature by detecting motion contrast from flowing blood cells, without the need for dye injection. While OCTA does provide three-dimensional volumetric data, it is more accurate to describe it as a non-invasive imaging technique that generates high-resolution, depth-resolved images of the retinal and choroidal vasculature. Please correct accordingly.

·      On page 3, line 93, please specify which commercial OCTA device was used in the study. Was it the RT-Vue Avanti? This is important for the study’s reproducibility.

·      Figures 1 and 2 are extremely small. Please include higher resolution images of 300 DPI or more.

·      For figure 2, you include outer retinal slab, FAZ, SCP, and CCBF, for week 1, 3 months, 6 months, and 12 months post-surgery. This figure could be improved by providing the same slab across multiple visits so that readers could appreciate the improvement in vascular density across the follow-up visits. In addition, the outer retinal slab image (a) does not provide much information here. Also, in (b), which plexus is this? Is this FAZ shown on the entire retinal slab? To simplify this, I suggest keeping the 4 post-operative date color fundus, and OCT images and use either all plexuses (outer retina, SCP, DCP, and CC) for each visit or just pick the superficial plexus for each visit.

·      Page 13, lines 332-333: Please consider rephrasing this sentence as follows: Apart from Michalewska et al.'s study, most of the research involving OCTA in ODP-M has been limited to case reports primarily focusing on OCTA-based assessments of the optic disc.

·      There are some grammatical errors throughout the manuscript. Please have the entire manuscript proofread once more to make sure these are taken care of. I’ll point out some of them here:

o   Page 1, line 25: increased the FAZ  increased FAZ.

o   Page 2, line 44: %25 to %75  25% to 75%

o   Page 2, line 48: which may leads  which may lead

o   Page 2, line 68: was the first to assess  is the first to assess

o   Page 3, line 90: strenght  strength

o   Table 3 and Table 4: SCP Total vascuaalar  SCP Total vascular

o   Page 11, line 259: decreased the FAZ  decreased FAZ

o   Page 12, line 269: may have related  may have been related

o   Page 13, line 356: In other work  In another work

Comments on the Quality of English Language

A few errors that need to be addressed. 

Reviewer 3 Report

Comments and Suggestions for Authors

The paper is interesting. The limitation is the low number of patients (8 cases) enrolled into trial.  I suggest the authors enroll more patients in the furthur study. 

Round 2

Reviewer 2 Report

Comments and Suggestions for Authors

Dear authors, thank you for addressing all the comments and suggestions raised in the review. Your detailed responses and the corresponding revisions have significantly improved the clarity, accuracy, and overall quality of the manuscript. The additional information you provided regarding segmentation errors, the definition of foveal, parafoveal, and perifoveal regions, and the specification of the commercial OCTA device used in the study have strengthened the methods section. The rephrasing of sentences and the correction of grammatical errors have enhanced the readability of the paper. Furthermore, the improvements made to Figures 1 and 2 have made the visual presentation of the data more effective. Your commitment to presenting a well-written and technically sound paper is commendable